# Comparing machine learning with case-control models to identify confirmed dengue cases

Tzong-Shiann Ho[1,2☯], Ting-Chia Weng[3,4☯], Jung-Der Wang[3,5,6], Hsieh-Cheng Han[7], Hao-Chien Cheng[8], Chun-Chieh Yang[8], Chih-Hen Yu[5], Yen-Jung Liu[8], Chien Hsiang Hu[9], Chun-Yu Huang[8], Ming-Hong Chen[9], Chwan-Chuen King[10]*, Yen-Jen Oyang[8]*, Ching-Chuan Liu[1,2]*

**1** Department of Pediatrics, National Cheng Kung University Hospital, College of Medicine, National Cheng Kung University, Tainan, Taiwan, Republic of China, **2** Center of Infectious Disease and Signaling Research, National Cheng Kung University, Tainan, Taiwan, Republic of China, **3** Department of Occupational and Environmental Medicine, National Cheng Kung University Hospital, Tainan, Taiwan, Republic of China, **4** Department of Family Medicine, National Cheng Kung University Hospital, Tainan, Taiwan, Republic of China, **5** Department of Internal Medicine, National Cheng Kung University Hospital, College of Medicine, National Cheng Kung University, Tainan, Taiwan, Republic of China, **6** Department of Public Heath, College of Medicine, National Cheng Kung University, Tainan, Taiwan, Republic of China, **7** Research Center for Applied Sciences, Academia Sinica, Taipei, Taiwan, Republic of China, **8** Institute of Biomedical Electronics and Bioinformatics, College of Electrical Engineering & Computer Science, National Taiwan University, Taipei, Taiwan, Republic of China, **9** Department of Medical Informatics, National Cheng Kung University Hospital, College of Medicine, National Cheng Kung University, Tainan, Taiwan, Republic of China, **10** Institute of Epidemiology and Preventive Medicine, College of Public Health, National Taiwan University, Taipei, Taiwan, Republic of China

☯ These authors contributed equally to this work.
* chwanchuen@gmail.com (C-CK); yjoyang@csie.ntu.edu.tw (Y-JO); liucc@mail.ncku.edu.tw (C-CL)

**Data Availability Statement:** All relevant data are full available within the manuscript and its Supporting Information files.

## Abstract

In recent decades, the global incidence of dengue has increased. Affected countries have responded with more effective surveillance strategies to detect outbreaks early, monitor the trends, and implement prevention and control measures. We have applied newly developed machine learning approaches to identify laboratory-confirmed dengue cases from 4,894 emergency department patients with dengue-like illness (DLI) who received laboratory tests. Among them, 60.11% (2942 cases) were confirmed to have dengue. Using just four input variables [age, body temperature, white blood cells counts (WBCs) and platelets], not only the state-of-the-art deep neural network (DNN) prediction models but also the conventional decision tree (DT) and logistic regression (LR) models delivered performances with receiver operating characteristic (ROC) curves areas under curves (AUCs) of the ranging from 83.75% to 85.87% [for DT, DNN and LR: 84.60% ± 0.03%, 85.87% ± 0.54%, 83.75% ± 0.17%, respectively]. Subgroup analyses found all the models were very sensitive particularly in the pre-epidemic period. Pre-peak sensitivities (<35 weeks) were 92.6%, 92.9%, and 93.1% in DT, DNN, and LR respectively. Adjusted odds ratios examined with LR for low WBCs [$\leq 3.2$ (x10$^3$/$\mu$L)], fever ($\geq$38˚C), low platelet counts [< 100 (x10$^3$/$\mu$L)], and elderly ($\geq$ 65 years) were 5.17 [95% confidence interval (CI): 3.96–6.76], 3.17 [95%CI: 2.74–3.66], 3.10 [95%CI: 2.44–3.94], and 1.77 [95%CI: 1.50–2.10], respectively. Our prediction models can readily be used in resource-poor countries where viral/serologic tests are inconvenient

**Funding:** The authors sincerely appreciate the financial support from the research grants of National Health Research Institutes (www.nhri.org.tw) (MR-108-GP-14 (CCK), NHRI-108A1-MRCO-0319191 (TSH)) and the Ministry of Science and Technology (www.most.gov.tw) (MOST-103-2314-B-006-009-MY3(TSH), MOST-107-2923-B-006-001(TSH), MOST-108-2923-B-006-001 (TSH)), Taiwan, which made this investigation possible. The funders had no role in study design, data collection and analysis, decision to publish, or preparation of the manuscript.

**Competing interests:** The authors have declared that no competing interests exist.

and can also be applied for real-time syndromic surveillance to monitor trends of dengue cases and even be integrated with mosquito/environment surveillance for early warning and immediate prevention/control measures. In other words, a local community hospital/clinic with an instrument of complete blood counts (including platelets) can provide a sentinel screening during outbreaks. In conclusion, the machine learning approach can facilitate medical and public health efforts to minimize the health threat of dengue epidemics. However, laboratory confirmation remains the primary goal of surveillance and outbreak investigation.

## Author summary

Identifying dengue cases early is crucial but challenging for healthcare professionals. This challenge is increased during large epidemics and is a particular problem in non-endemic areas with limited experienced staff. To improve dengue diagnosis, we investigated how to exploit machine learning (ML)-based prediction models and identified four key variables [age, fever, white blood cell counts (WBCs), and platelet counts], which are compatible with clinical and epidemiological knowledge. With these variables, the ML prediction models [decision tree (DT), deep neural network (DNN)] and the logistic regression model developed for identifying laboratory-confirmed dengue cases produced areas under curve (AUCs) of the receiver operating characteristic (ROC) curves ranging from 83.75% to 85.87%. This implies that the prediction models may serve as a pivotal component of an integrated dengue surveillance system and they required only a single complete blood count (CBC) examination. The sensitivities, positive prediction values, and accuracies for major risk factors in the two machine learning models were close to those of the regression models. For future applications, the DNN models with superior performance can be employed at epidemic sites with adequate computer facilities, while the DT and regression models with interpretable prediction logic can be employed at sites with limited or no computer facilities. Artificial intelligence and clinical parameters identified from this study may aid when laboratories are overwhelmed, but should never replace laboratory confirmation.

## Introduction

Outbreaks of dengue have continuously increased worldwide in recent decades [1, 2], while global warming and extreme weather conditions have worsened [3]. Dengue is the most influential arbovirus disease in the world, according to global morbidities and mortalities [4, 5]. To reduce the magnitude of dengue epidemics and to decrease fatalities, early detection of dengue cases through surveillance to target high risk areas and populations has become one of the most important public health strategies in many countries [6, 7]. However, the infection of dengue virus (DENV) results in a wide clinical spectrum of symptoms, ranging from subclinical infection, to mild dengue fever (DF), to severe dengue [8, 9]. Under-reporting or late recognition of dengue is frequent when patients present atypical symptoms/signs, including undifferentiated fever, gastrointestinal syndrome, and influenza-like illness, particularly in children or patients at the febrile phase or at the early stage of epidemics [10, 11]. In the febrile phase, dengue patients usually present non-specific symptoms/signs or viral syndrome when they first visit primary care physicians [8]. At the population level, dynamic changes of clinical

manifestations have occurred from early to middle and late stages of the same epidemic [10]. Therefore, relying only on clinical surveillance of dengue, using the definitions of suspected or probable dengue cases may jeopardize resource allocations during large-scale epidemics.

As dengue epidemics have become more and more severe globally over the years [12], epidemiological studies in Taiwan have demonstrated that epidemic severity increased from early to middle and late stages of the same epidemics [13, 14]. Cuba also reported similar findings [15]. In other words, promptly recognizing and monitoring dengue cases from beginning of epidemics, enabling immediate implementation of prevention and control measures is necessary to minimize epidemic severity. Unfortunately, most problems of dengue surveillance have continued with little improvement. Major problems of global surveillance of dengue include the following: (1) passive surveillance hinders accurate information of total dengue cases [10], (2) many reported dengue cases were clinically defined rather than laboratory-confirmed [6, 16], and (3) under-estimates of mild dengue cases frequently occur when more severe or fatal dengue cases appear [17]. Accordingly, how to accurately predict laboratory-confirmed dengue cases in areas with limited resources is a major challenge for public health decision-makers. In this study, we addressed this challenge by conducting comprehensive analyses on how prediction models built with different types of machine learning algorithms and different variable sets performed in identifying laboratory-confirmed dengue cases among those patients with dengue-like illness (DLI). One of the most significant findings was that the prediction models built with only 4 key input variables, being age, body temperature, count of white blood cells (WBC), and count of platelets (PLT), were able to deliver the same level of performance as the prediction models built by incorporating additional 14 variables, including gender, hemoglobin level, patient's triage levels at ED, vital signs, and comorbidities. This result is in conformity with clinical knowledge as well as epidemiological characteristics and implies that the prediction models can serve as a pivotal component of an integrated dengue surveillance system by requiring only a single complete blood count (CBC) exam. The level of performance observed in our experiments further implies that these prediction models built with only 4 input variables can be employed to provide real-time syndromic surveillance in areas without adequate medical resources and access to viral/serologic tests. On the other hand, in areas with adequate medical resources, the prediction models can serve as complementary tools to raise the sensitivity of an integrated surveillance system.

In fact, exploiting machine learning algorithms and statistical methods to facilitate dengue diagnosis has been studied by scientists around the world for over 10 years [18–22]. In recent years, researchers started to exploit more advanced machine learning algorithms such as the Bayesian network [23]. All of these previous studies focused on how to predict dengue diagnoses, dengue phenotypes, or high-risk groups of severe illness and/or mortality but did not address how to effectively exploit alternative machine learning algorithms with very different application values. In this respect, it is particularly of interest to compare the performance delivered by the state-of-the-art deep neural networks (DNN) models [24, 25] and the conventional decision tree (DT) models [26]. It is generally observed that the DNN models can deliver superior performance compared to alternative machine learning algorithms [27] but it is almost impossible for a user to figure out how a prediction is made [28]. On the other hand, the DT models are favored in many applications due to the explicit prediction logic output by the DT algorithm. However, it is also well known that the prediction performance of the DT models generally cannot match that of the prediction models built with advanced machine learning algorithms such as the DNN, the support vector machine (SVM) [29], and the random forest [30]. In this respect, Flaxman and Vos concluded their experiences and proposed "using an explainable approach, even with a reduction in accuracy, can be superior" [29]. The third approach investigated in this study was the conventional LR, which can output the crude

and adjusted odds ratio (OR) for each input variable [31] and has been frequently applied in epidemiologic studies with a case-control design. It is of interest to learn how the machine learning based prediction models and the conventional statistics-based models compare.

## Methods

### Study population

An unprecedented dengue epidemic occurred in Taiwan during 2015 and resulted in 22,777 laboratory-confirmed cases [32]. S1 Fig shows the epidemic curve. With the data collected during this epidemic, we then generated the dataset used in this study, which contained dengue-like illness (DLI) cases admitted to the emergency department (ED) from January 1 to December 31, 2015 (the epidemic year) at National Cheng Kung University Hospital (NCKUH) in Tainan City in southern Taiwan. All the clinical diagnoses of DLI were made by clinicians according to the 1997 or 2009 WHO clinical definition of probable dengue [9]. By these definitions, a patient was diagnosed to suffer DLI and coded with corresponding ICD codes, if the patient had fever along with any two of the following clinical features: nausea/vomiting, rash, aches and pains, tourniquet test positive or any warning signs. In total, there were 100,491 visits to the ED of NCKUH (NCKUH-ED) during 2015. Among them, 3698 patients canceled the emergency consultation and therefore were excluded. Furthermore, 6611 patients were re-admitted to the ED of the NCKUH within 36 hours and therefore their records were merged. Our analyses showed that these excluded cases and merged cases were evenly distributed by months. In other words, the numbers of excluded cases and merged cases were not affected by the dengue endemic. Fig 1 illustrates the procedure employed to generate the dataset used in our analyses. Among the 100,491 patients admitted to NCKUH-ED in 2015, 6,368 patients (6.34%) met our definition of a DLI case, given that (1) the patient was coded with ICD-9 061 (dengue), 0654 (mosquito-borne hemorrhagic fever), 0663 (other mosquito-borne fever), or v735 (screening examination for other arthropod-borne viral diseases) for dengue fever; or (2) the patient received one or more dengue serological and/or virological tests, including dengue NS1, dengue-IgM, viral load of DENV, or dengue serotyping using polymerase chain reaction (PCR) to detect DENV-1 and DENV-2. We then excluded those 1,302 DLI cases (excluded group) who did not receive a dengue laboratory test and another 172 DLI cases due to missing values in any of the 18 variables incorporated in our analysis, which include age, gender, patient's triage levels at ED, and the blood counts (CBC), vital signs, and comorbidities listed in Table 1. In the end, we had 4,894 DLI cases included in our dataset with 2,942(60.12%) laboratory-confirmed dengue cases and 1,952 non-dengue control cases. The characteristics of these two groups (confirmed dengue cases and the controls) are summarized in Table 1. Meanwhile, the characteristics of the included group and the excluded group are summarized in S1 Table.

### Variable selection

Our variable selection process began with identifying features at the initial clinical presentation that might provide crucial information to assist in diagnosing laboratory-confirmed dengue cases. Based on physicians' medical knowledge and clinical experience, 18 variables were initially identified, including age, gender, and the data from complete blood count (CBC), patient's triage levels at ED, vital signs, and comorbidities. We then employed the cutoff values shown in S2 Table to stratify these variables and computed the crude odds ratios of the 18 variables. The cutoff values represented the normal ranges of the tests (i.e. serving as reference values) which have been routinely used in the NCKU Hospital with greater differentiation of "normal" versus "abnormal" (high or low). Based on the crude odds ratios shown in S3 Table, we identified the following four key variables: age, body temperature, counts of WBCs and

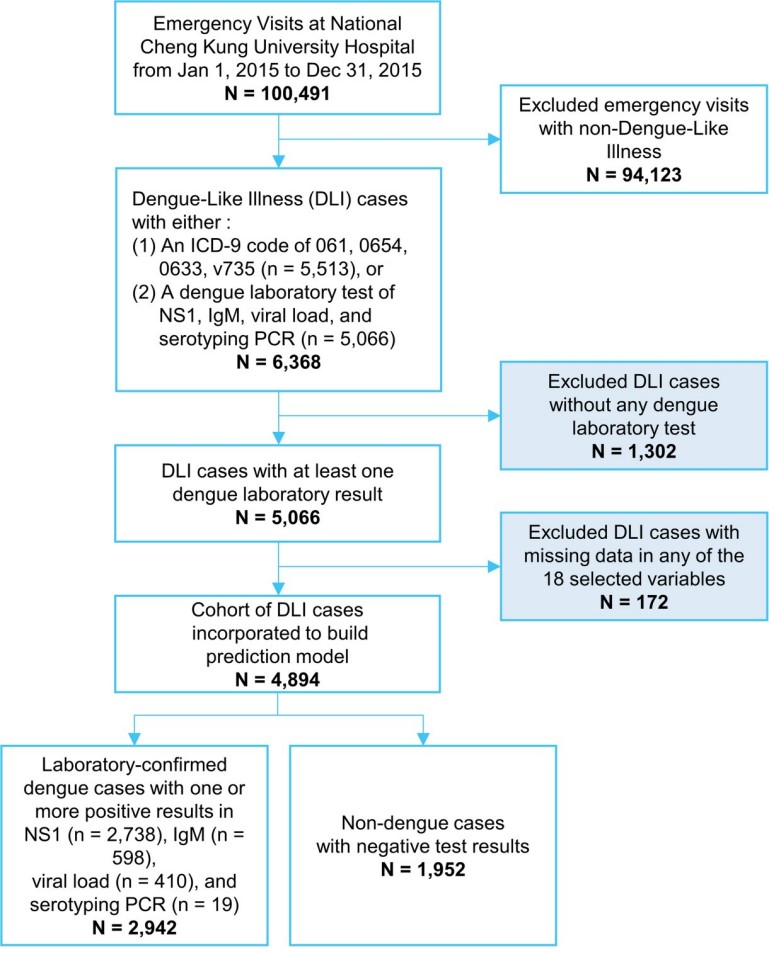

**Fig 1. Flow diagram for extracting 2942 laboratory-confirmed dengue cases (case group) and 1952 non-dengue cases (control group) from source of study population.**

platelets. Aiming to evaluate whether these four key variables essentially provide all the crucial information for identifying confirmed dengue cases, we further included hemoglobin (Hb) and gender to form a six-variable feature set based on physicians' suggestions and epidemiological characteristics from past findings. The adjusted odds ratios shown in Table 2 confirmed the robustness of the four key variables.

## Prediction models

Aiming to evaluate the effectiveness of exploiting the key variables identified above to predict laboratory-confirmed dengue cases from patients with DLI, we developed three types of prediction models, two types of machine learning models, namely, the decision tree (DT) models [26] and deep neural network (DNN) models [24, 25], and logistic regression (LR) models [31]. The performance of the DT models was of interest due to its ease of interpretability, a unique feature favored by many physicians. However, the algorithm for building a DT model is based on univariate analysis and does not incorporate any linear or non-linear transformation. As a result, the prediction performance of the DT models may not match other types of prediction models when applied to cases in which samples with different labels are separated by non-linear boundaries. In this respect, due to the complicated non-linear transformations

**Table 1. Demographic and clinical characteristics of those included study subjects (laboratory-confirmed dengue and non-dengue cases) and excluded ED patients at NCKU Hospital, Jan. 1 to Dec. 31, 2015.**

| | Laboratory-confirmed Dengue Cases | Non-Dengue Patients (Control Group) | p-value* | Excluded |
|---|---|---|---|---|
| **Emergency Visits** | 2942 | 1952 | | 1474 |
| **Age (years)** | | | | |
| Age < 18 | 174 (5.91%) | 183 (9.38%) | <0.001 | 161 (10.92%) |
| 18 ≤ age < 65 | 1871 (63.6%) | 1382 (70.8%) | | 935 (63.43%) |
| 65 ≤ age | 897 (30.49%) | 387 (19.83%) | | 378 (25.64%) |
| **Age (mean ± SD)** | 50.24± 21.47 | 41.68 ±23.14 | <0.001 | 45.8±22.58 |
| **Gender** | | | | |
| Female | 1473 (50.07%) | 945 (48.41%) | 0.2565 | 783 (53.12%) |
| Male | 1469 (49.93%) | 1007 (51.59%) | | 691 (46.88%) |
| **Epidemic periods** | | | | |
| Pre-peak: wks ≤ 35 | 399 (13.56%) | 150 (7.68%) | <0.001 | 361 (24.49%) |
| Peak: 35 < wks ≤ 40 | 1912 (64.99%) | 1077 (55.17%) | <0.001 | 810 (54.95%) |
| Post-peak: 40 < wks | 631 (21.45%) | 725 (37.14%) | <0.001 | 303 (20.56%) |
| **Severity** | | | | |
| Non-Hospitalized | 2443 (83.04%) | 1495 (76.59%) | <0.001 | 1148 (77.88%) |
| Hospitalized | 420 (14.28%) | 389 (20.08%) | <0.001 | 270 (18.32%) |
| ICU | 36 (1.22%) | 29 (1.49%) | 0.5115 | 24 (1.63%) |
| Death | 43 (1.46%) | 36 (1.84%) | 0.2983 | 32 (2.17%) |
| **Triage Vital Signs** | (mean±SD) | (mean±SD) | | |
| Temperature (˚C) | 38.33±0.98 | 37.97±0.99 | <0.001 | 37.64±1.43 |
| Systolic BP (mmHg) | 135±22 | 133±22 | <0.001 | 127±21 |
| Diastolic BP (mmHg) | 82±15 | 82±15 | 0.4068 | 79±20 |
| Heart Rate (BPM) | 100±19 | 102±20 | 0.0039 | 92±20 |
| Respiratory Rate (/min) | 20±3 | 20±3 | 0.0001 | 20±2 |
| **Blood Counts** | (mean±SD) | (mean±SD) | | |
| WBCs ($10^3/\mu L$) | 5.25±2.70 | 9.18±4.29 | <0.001 | 4.81±3.33 |
| Platelets ($10^3/\mu L$) | 148.08±65.85 | 205.79±76.48 | <0.001 | 114.63±79.80 |
| Hemoglobin (g/dL) | 13.39±1.71 | 13.03±2.07 | <0.001 | 13.57±1.93 |
| **Comorbidities** | | | | |
| Heart Disease | 332 (11.44%) | 213 (11.06%) | 0.6850 | 153 (10.38%) |
| CVA | 147 (5.06%) | 118(6.13%) | 0.1122 | 55 (3.37%) |
| CKD | 653 (22.49%) | 436 (22.64%) | 0.9069 | 288 (10.54%) |
| Severe Liver Disease | 250 (8.61%) | 185 (9.61%) | 0.2376 | 144 (9.77%) |
| DM | 532 (18.33%) | 348 (18.07%) | 0.8206 | 253 (17.16%) |
| Hypertension | 584 (20.12%) | 354 (18.38%) | 0.1352 | 264 (17.91%) |
| Cancer | 528 (17.95%) | 398 (20.39%) | 0.0327 | 208 (14.11%) |

Pre-peak: Before Epidemic Peak in the Epidemic Curve; SD: Standard Deviation; ICU: Intensive Care Units; BP: Blood Pressure; BPM: Heart Rate as Beats per Minute; WBCs: White Blood Cells; CVA: cerebral vascular accident; CKD: Chronic Kidney Disease; DM: Diabetes Mellitus

involved, the state-of-the-art DNN models generally can produce superior prediction performance in comparison with other types of prediction models [27]. However, the DNN based models generally contain a large quantity of coefficients and therefore it is almost impossible for a user to figure out how the prediction model works. In this study, we further investigated how the conventional LR models performed because logistic regression is widely exploited in medical research and epidemiology, and many physicians are familiar with its mathematical

**Table 2. The Crude and Adjusted odds ratios for both 4-variable set and 6-variable set.**

| Variables | | 4-variable set | | 6-variable set | |
|---|---|---|---|---|---|
| | Crude OR | Adjusted OR | 95% C.I. | Adjusted OR | 95% C.I. |
| Young/Adult | 0.70 | 0.59 | (0.46,0.76) | 0.65 | (0.50,0.84) |
| Elder/Adult | 1.71 | 1.77 | (1.50,2.10) | 2.19 | (1.84,2.62) |
| Fever | 1.92 | 3.17 | (2.74,3.66) | 3.28 | (2.83,3.79) |
| Low_PLTs | 3.95 | 3.10 | (2.44,3.94) | 3.03 | (2.38,3.87) |
| Low_WBCs | 4.49 | 5.17 | (3.96,6.76) | 5.41 | (4.13,7.10) |
| High_WBCs | 0.10 | 0.08 | | 0.08 | (0.07,0.10) |
| Male/Female | 0.94 | | | 1.22 | (1.11,1.41) |
| Low_Hb | 0.67 | | | 0.49 | (0.42,0.58) |
| High_Hb | 1.13 | | | 1.13 | (0.86,1.49) |

OR: Odds Ratios; 95% CI: 95% Confidence Intervals in parentheses

4-variable set: Age, Body Temperature, Counts of white blood cells (WBCs) and platelets (PLTs)

6-variable set: Age, Body Temperature, Counts of WBCs, PLTs, and hemoglobin (Hb), and Gender

fundamentals. S4 Table summarizes the software packages employed to build the DT and LR models and the main characteristics of the DNN models. With respect to the structure of the DNN models, we actually investigated the performance of more complicated networks and observed that the simple network structure shown in S4 Table delivered the same level of performance in comparison with more complicated network structures. In this experiment, we set the dimension of the network to either 16 or 64 and the number of layers to 3, 10, or 100. The last issue with respect to building the prediction models was how the distributions of the dataset should be handled. Since the dataset contained 2,942(~60%) positive subjects and 1,952(~40%) negative subjects, we did not employ any procedure to address this issue. This issue is of concern only if the numbers of subjects in different groups, e.g. positive and negative, are highly unbalanced.

## Performance evaluation procedures

In this study, the performance delivered by the three types of prediction models addressed above, i.e. the DT, DNN, and LR models, was evaluated based on the area under the receiver operating characteristic (ROC) curve [33], which is commonly referred to as the area under curve (AUC). In the following discussion, we will elaborate on the procedures employed to obtain the ROC curve for the DT models and then describe the procedures employed for the DNN and the LR models. In order to obtain the receiver operating characteristic (ROC) curve for the DT models, we set parameter prior to alternative values between 0 and 1. For each setting of parameter prior, we carried out the 10-fold cross validation procedure [24] shown in Fig 2 to evaluate the performance characteristics of the DT models with this particular setting. During each iteration of the 10-fold cross validation procedure, a DT model was generated. For example, S2 Fig shows one of the DT models generated during the 10-fold cross validation procedure with prior set to 0.388. Assuming that this particular DT model was generated in the k-th iteration of the 10-fold cross validation procedure, the performance characteristics of this particular DT model was evaluated by feeding the k-th subset into the model. For this particular DT model, the following performance data was obtained: sensitivity = 90.1%, specificity = 63.6%, PPV = 78.9%, NPV = 81.0%, and accuracy 79.6%. S3 Fig shows a DT model generated with prior set to 0.636. For this particular DT model, the following performance data was obtained: sensitivity = 66.3%, specificity = 80.5%, PPV = 83.7%, NPV = 61.3%, and

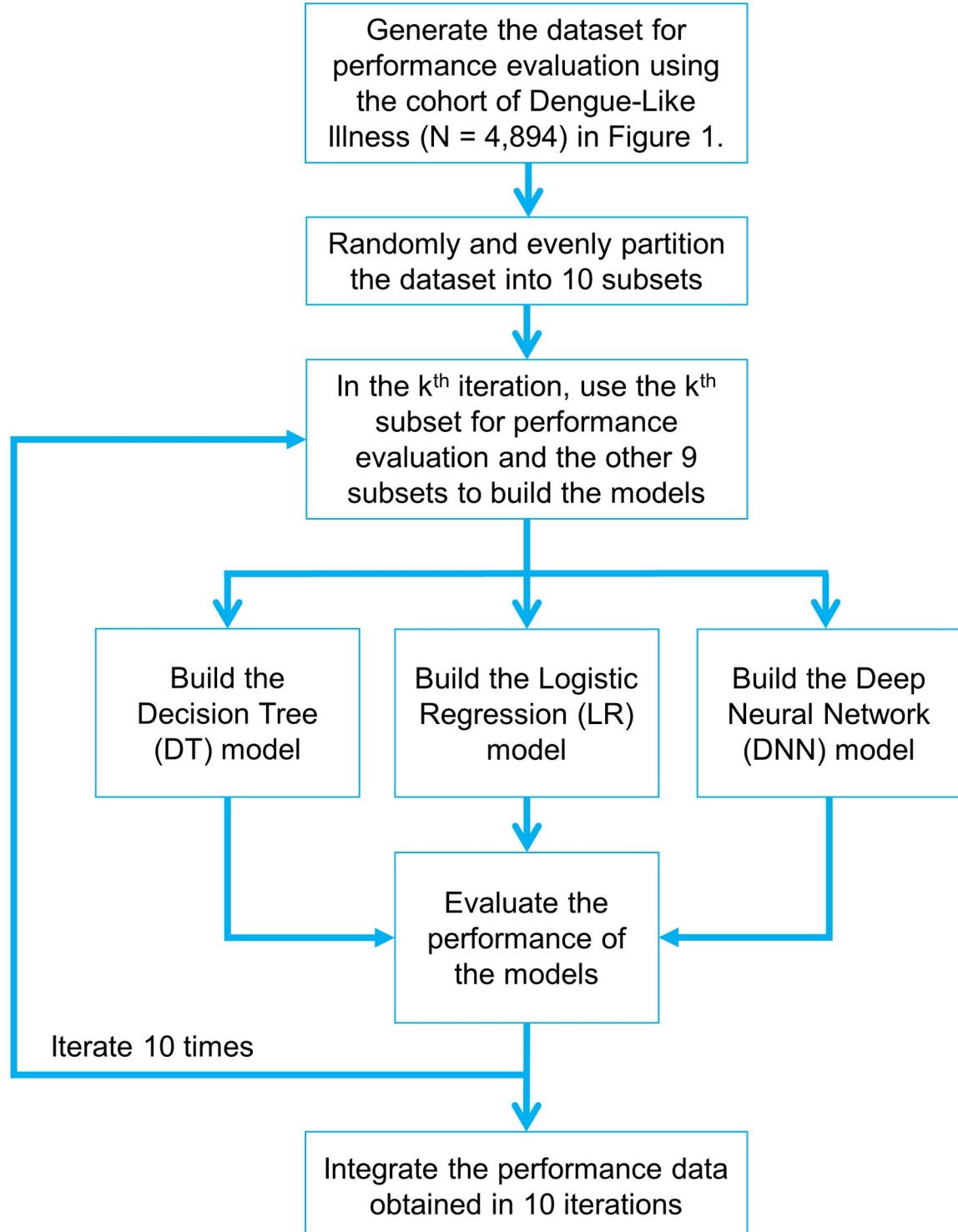

**Fig 2. Performance evaluation procedure based on 10-fold cross validation.** In each iteration of the 10-fold cross validation procedure, 90% of the patients' records in the cohort were used to build the prediction model. Then, the remaining 10% of the patients' records without the end results were fed into the prediction model and the predictions made by the prediction model were compared with the end results recorded in the cohort to evaluate how accurate the prediction model performed. The iteration was repeated 10 times with each of the 10 subsets being used for performance evaluation once and only once [24].

accuracy = 72.0%. For each specific setting of parameter prior, we repeated the 10-fold cross validation procedure 20 times and computed the means and standard deviations of the main performance metrics. Finally, with all the performance data collected by setting parameter prior to alternative values, we drew the ROC curves accordingly.

For generating the ROCs with the DNN and logistic regression models, we followed a similar procedure to obtain the performance readings, with the exception being that the parameters involved were the thresholds employed to convert numerical outputs to predicted categorical outcomes.

### World Health Organization (WHO) clinical definition of dengue

The clinical diagnosis of dengue-like illness in Taiwan was usually made according to the 1997 or 2009 WHO clinical definitions. In light of epidemiological or laboratory evidence supporting a dengue virus infection, the 1997 WHO clinical definition of dengue was defined as fever with two of the following clinical features: headache, arthralgia, retro-orbital pain, rash, myalgia, hemorrhagic manifestation or leukopenia [8]. On the other hand, the 2009 WHO clinical definition of probable dengue was defined as fever with two of the following clinical features: nausea/vomiting, rash, aches and pains, tourniquet test positive or any warning signs [9]. The reported sensitivity and specificity of the 1997 and 2009 WHO definitions in predicting dengue [34] were also presented in the Fig 3 for better comparison.

### Data validation

To ensure data accuracy, we independently repeated all the experiments presented in this article at least two times. The results of AUCs, sensitivities, specificities, PPVs, and accuracies from the two independent runs were very close. All original dataset and software codes will be made available upon requests.

### Ethics statement

This study was approved by the Institutional Review Board (IRB) of National Cheng Kung University Hospital (NCKUH-IRB Approval Number: A-ER-108-209). Data were fully de-identified and anonymized to protect participants' privacy, and only aggregated data were used for further analyses and statistical tests.

## Results

### Demographic analyses

Among 6,368 patients with dengue-like illness (DLI) admitted to the emergency department (ED) of NCKUH in 2015, 2,942 cases (46.20%) were confirmed to have dengue due to one or more positive results with dengue-NS1, IgM, PCR, or viral load of DENV tests, i.e. the "confirmed dengue group". The "control group" comprised 1,952 cases with dengue-negative results from laboratory tests. The remaining 1,474 cases were excluded from our dataset due to either no laboratory results or missing values on one or more of the 18 variables in our initial variable set. The demographic characteristics and clinical features of the confirmed dengue cases group and the control group are summarized in Table 1. The confirmed dengue case group was distinctive from the control group by the following characteristics: (1) significantly older, (2) less likely to be hospitalized, (3) significantly higher mean of body temperature, (4) significantly lower mean counts of white blood cells (WBCs) and platelets, and (5) higher hemoglobin. Meanwhile, no major difference was observed with respect to gender distributions and proportions of the patients who suffered from the following six comorbidities: heart

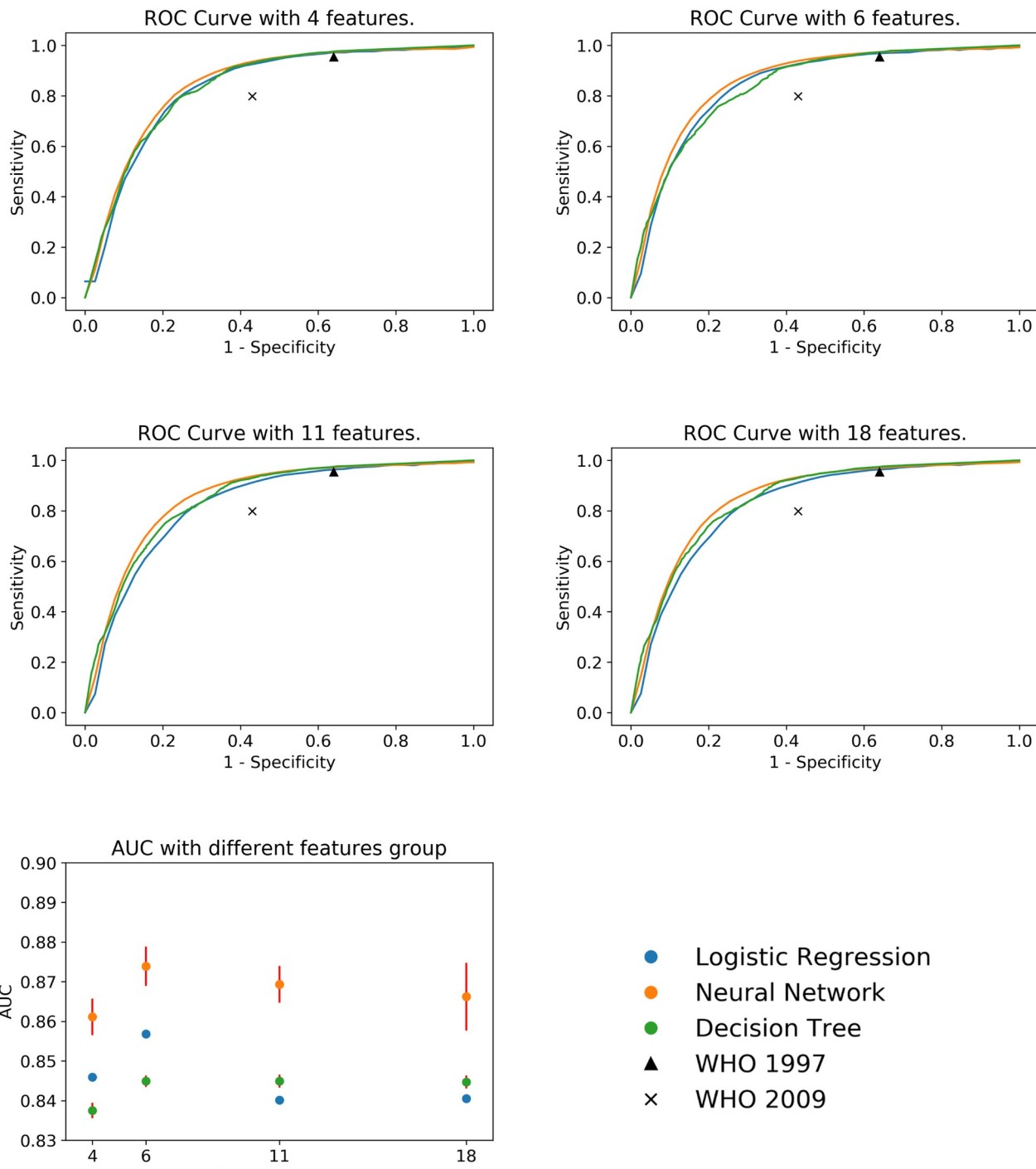

**Fig 3. Performance delivered by two machine learning methods (decision tree, deep neural network) and logistic regression models with 4, 6, 11, and 18 input variables.** AUCs with 4 input variables- DT: 83.75%±0.17%, DNN: 85.87%±0.54%, LR: 84.60%±0.03%; AUCs with 6 input variables-DT: 84.49%±0.11%, DNN: 86.95%±0.45%, LR: 85.69%±0.09%; AUCs with 11 input variables- DT: 84.49%±0.14%, DNN: 86.40%±0.64%, LR: 84.04%±0.07%; AUCs with 18 input variables- DT: 84.47%±0.14%, DNN: 86.35%±0.63%, LR: 84.07%±0.07%. The reported sensitivities/specificities for determining dengue, based on the 1997 and 2009 definitions were 95.4%/36.0% and 79.9%/57.0%, respectively [34].

disease, cerebrovascular disease, and chronic kidney disease, cirrhosis of the liver, diabetes, and hypertension. Nevertheless, a lower percentage of the patients in the confirmed dengue cases group suffered cancer than the patients in the control group but the difference is marginal.

## Performance evaluation

Fig 3 summaries how the DT, DNN, and logistic regression (LR) models performed with four different variable sets, with 4, 6, 11 and 18 variables, respectively. The smallest variable set included only the four key variables identified based on the analyses of crude odds ratios (S3 Table) and adjusted odds ratios (Table 2). The six-variable set was derived from the four-variable set by adding gender and count of Hb. The 11-variable set was derived from the six-variable set by adding the five vital sign features in our initial variable set. Finally, we incorporated the initial 18-variable set involving seven comorbidity features. The overall performance of the DT, the DNN, and the LR models with different variable sets is shown by the receiver operating characteristic (ROC) curves in Fig 3(A), 3(B), 3(C), and 3(D). Fig 3(E) summarizes the areas under the ROC curves (AUCs) shown in Fig 3(A), 3(B), 3(C) and 3(D). Two interesting observations deserve our attention. Firstly, all the AUC numbers shown in these figures were close to or above 84% and the DNN models marginally outperformed the DT models and the LR models. Secondly, incorporating more input variables did not necessarily lead to better performance. In fact, for all three types of models, the performance differences due to incorporating different variable sets are marginal. This observation implies that the four key variables, i.e. age, body temperature, counts of WBCs and PLTs, together provide essentially all the information available at the initial clinical presentations for identifying confirmed dengue cases.

## Subgroup analyses of confirmed dengue cases

For future applications of our prediction models, we conducted comprehensive analyses on how these models performed with specific patient groups. We partitioned patients based on age, gender, and major epidemiological characteristics. S5 Table summarizes the performance of the DT, DNN and LR models that delivered sensitivity at the 90% level with four key variables. The detailed performance data with the three prediction models for DT, DNN, and LR are shown in S6, S7 and S8 Tables, respectively. The first interesting observation is that all prediction models delivered the highest level of sensitivity when applied to predicting patients admitted during the pre-peak period. This is a favorable characteristic as detecting early cases is crucial for prevention and control of dengue. In the meantime, all these models delivered a higher level of specificity when applied to predicting those patients who were admitted during the peak and the post-peak periods. Again, this is favorable, as high-specificity screening mechanisms are desirable once outbreak occurs. The second interesting observation is that for pediatric patients the DT model delivered higher sensitivity but lower specificity (95.5% sensitivity and 54.8% specificity) than the DNN model, which delivered 87.1% sensitivity and 73.5% specificity.

## Discussion

The spreading of DENV has been expanding in recent years [4, 6, 7]. Global epidemiology of dengue shows that the time interval between regional epidemics after World War II have become shorter particularly in urban centers of Southeast Asian countries where dengue is endemic [35, 36]. As a result, dengue has become a continuing global threat that may cause a great loss of human life and a great impact on social welfare [37, 38]. In particular, large-scale

or unanticipated epidemics of dengue often overwhelm healthcare systems [39, 40] and lead to a large number of severe and fatal cases [39].

With such a great challenge, effective and efficient surveillance of dengue is essential for timely detecting outbreaks early on, monitoring the trend of incidence, and evaluating prevention and control measures [41]. If dengue cases are not detected early, continuous presence of DENV in the community will result in selection of the virus strains or subvariants with increasing percentages of severe cases occurring in later stages of the epidemic [13, 14, 42]. However, many primary health care professionals may not be familiar with important clinical features of dengue [43]. Furthermore, many infectious diseases have dengue-like nonspecific symptoms/signs [44]. Therefore, rapid detection of laboratory-confirmed dengue cases is crucial in precise targeting for early intervention with better resource allocation. Early laboratory diagnosis of DENV, which can assist in clinical case management and public health planning, has many limitations. Three widely used approaches are costly [45, 46], including molecular diagnosis of viral nucleic acid, antigen of non-structure protein 1 (NS1), and human antibody. The PCR tests to detect DENV-RNA, is not suitable for patients who seek medical care late and it is not feasible in areas with limited resources [47]. On the other hand, detection of DENV-NS1 antigen is fast [48] but patients with secondary DENV infection show an earlier decrease in NS1 levels [49]. Serological tests of DENV-IgM and IgG antibodies have to consider cross-reactivities of other flaviviruses [50] and the timing of specimen taking [45, 46]. Therefore, laboratory diagnosis is time-consuming and requires expertise and tests with high sensitivity and specificity [51], all of which usually limits its availability at local clinics or small hospitals. With all these observations in mind, we resorted to machine learning approaches to facilitate screening patients for dengue diseases.

In this study, we conducted comprehensive analyses to evaluate how prediction models built with different types of machine learning algorithms and different variable sets performed in identifying laboratory-confirmed dengue cases among those patients with dengue-like illness (DLI). In fact, as mentioned earlier, exploiting machine learning algorithms and statistical methods to facilitate dengue diagnosis has been studied by scientists around the world for over 10 years [18–22] and researchers have started to exploit more advanced machine learning algorithms in recent years [23]. Nevertheless, all these previous studies did not focus on how to effectively exploit alternative machine learning algorithms with very different application values. For example, Potts and et. al. employed the decision tree algorithm to predict those pediatric patients who were likely to suffer from severe symptoms [21]. In this respect, it is particularly of interest to learn how the state-of-the-art DNN models perform in comparison with the prediction models built with the conventional DT and LR algorithms. It is generally observed that due to the multiple layers of non-linear transformations involved the DNN models can deliver superior performance to the prediction models built with the other machine learning approaches. However, the complicated non-linear transformations involved also make it almost impossible for a user to figure out the decision rules that the DNN model follows to make predictions [24]. Since for some medical and public health applications, the decision rules followed by the machine learning based prediction models can provide valuable insights, Flaxman and Vos concluded their experiences and proposed "using an explainable approach, even with a reduction in accuracy, can be superior" [28]. Accordingly, it is of interest to evaluate the performance of the DT models because the DT models are favored for many applications due to the explicit prediction logic output by the DT algorithm. However, due to lack of linear or non-linear transformations involved the prediction performance of the DT models generally cannot match that of the prediction models built with advanced machine learning algorithms such as the DNN, the support vector machine (SVM) [29], and the random forest [30]. The third approach investigated in this study was the conventional LR, which

can output the crude and adjusted odds ratio (OR) for each input variable [31] and has been frequently exploited in epidemiologic studies with a case-control design. The results in Fig 3 reveal that the three different types of prediction models investigated in our study basically delivered the same level of performance with the DNN models slightly outperforming the DT models and the LR models.

Another major finding of this study was that with only four key input variables, not only DNN prediction models but also conventional DT models were able to provide performance required for clinical applications. In particular, both the DT and DNN models with overall sensitivities at 90% delivered higher sensitivities, 92.6% and 92.8%, respectively, when applied to identifying laboratory-confirmed dengue cases in the pre-epidemic period than in other epidemic periods. This observation implies that the machine learning based prediction models can be exploited in the pre-epidemic stage to provide medical practitioners with a real data based objective diagnosis utility to complement clinical judgment solely based on personal experiences. From a public health viewpoint, our high-sensitivity models can be an effective surveillance tool in the pre-epidemic period. Once the number of cases dramatically climbs during the peak and post-peak periods, prediction models with high specificity can be exploited to identify laboratory-confirmed dengue cases. In this respect, our prediction models with overall 80% specificity delivered reasonably good sensitivities ranging from 69.7% to 79.9%. Notably, the four key input variables identified in this study (age, body temperature, and counts of both WBCs, and platelets) can be easily collected with minimal cost. Therefore, the prediction models developed here can be widely exploited at outbreak sites for real-time monitoring of epidemic trends. At sites with adequate computer facilities, the DNN models can be applied to achieve the highest prediction performance. On the other hand, at sites with very limited or even no computer resources, the DT models or the explicit prediction logic of the DT models alone can be used to obtain reasonable prediction performance.

With respect to practical applications of machine learning based prediction models, the computer resources available impose a major concern. Both the DT and LR algorithms can be efficiently executed on a typical personal computer. On the other hand, due to the nature of the back-propagation algorithm [52], which is the prevailing approach to train a DNN model and involves a lot of array processing, a computer equipped with a graphic processing unit (GPU) is normally required to carry out the training efficiently. Furthermore, if we add more layers to a DNN structure, then the training time will increase dramatically. Therefore, a simpler DNN structure is favored, if it can deliver the same level of prediction performance as a more complicated structure. In this respect, we evaluated the prediction performance delivered by more complicated DNN structure with 10 and 100 layers and observed no significant performance difference in comparison with the simple structure with only 3 layers (S4 Table). Although training a DNN model requires special hardware, once the training process is completed, the DNN model can be executed on a typical personal computer efficiently. This implies that the training process of a DNN model can be executed in a centralized computer facility and then the model can be distributed to local clinics equipped with minimal computer hardware.

One of our unique findings is to link the two machine learning approaches and conventional method of LR. In this study, we simultaneously collected subjects as cases and controls, and the collection procedure was unrelated to any exposure or risk factors, which could be regarded as one form of density sampling so that the odds ratios in Table 2 can be interpreted as a rate ratio [53, 54] for major risk factors of cases, and this finding is consistent with current understanding of dengue pathophysiology and epidemiology. In other words, an LR model based on domain knowledge on dengue epidemiology could be used to corroborate the results and interpretation of machine learning models of DT and DNN algorithms. The adjusted

odds ratios from the LR model from high to low are quite similar to the ranking of variables located from top to bottom in the DT model. In other words, DT can verify the selected variables from the DNN model and LR can further verify the DT results by ranking the adjusted ORs. Moreover, LR models can be constructed on personal computers or laptop at low cost. Once the prediction model is built at central lab, we may still utilize the models in remote areas through the internet and cloud technology.

An ideal dengue test should distinguish dengue from other infectious diseases, be highly sensitive, easy to use, inexpensive, rapid in getting results, and have stable reagents which are stable at temperatures above 30°C for usage in settings with limited or no optimal storage options [55]. The four input variables identified in this study to predict confirmed dengue cases meet all these criteria. Although many prediction models of dengue have focused on trends in incidence of dengue [56] or severe dengue cases [57, 58], very few studies predicted confirmed dengue cases. Among the relevant clinical variables, leucopenia, thrombocytopenia, elevated aminotransferases, low C-reactive protein (CRP) and prolonged activated partial thromboplastin time (aPTT), were useful predictive markers for early diagnosis of dengue during the 2007 DENV-1 outbreak in Tainan [59]. However, data of CRP and aPTT may not be available in primary health care settings. In fact, our four predictors have been frequently used in clinical risk scores for adult dengue cases [57]. In Singapore, the best decision tree prediction for hospitalized adult patients with DHF included a history of clinical bleeding, serum urea, and serum total protein but that model offered positive predictive value of 7.5%, and accuracy of 48.1% [58], and both serum urea and total protein may not be frequently measured. Comparing all those findings, our four input variables involving fever (body temperature≧38°C), numbers of WBCs and platelets, and age, which are consistent with clinical observations of dengue [8, 9], are most feasible for wide application. Low WBC count and low platelet count, are important clinical parameters for suspicion of dengue [9]. Moreover, age is an important risk factor because most of Taiwan's fatal dengue cases in 2015 were elderly [32]. In other words, measuring body temperature and a single CBC tube, plus age in a dengue non-endemic area like Taiwan can be employed for clinical surveillance in real time to assess where high-risk areas and populations are. Furthermore, our novel DNN algorithm approach employing an advanced machine learning method also verified the four variables highly selected by the DT algorithm, and this DNN approach is much simpler, without requiring many variables of symptoms/signs as each of most dengue clinical guidelines. We also built prediction models with larger sets of input variables, but found no statistical performance difference.

This study has seven major limitations. First, most Taiwan dengue patients are adults, and are caused by a predominant, single serotype of DENV that is different from dengue-endemic countries, where cases are mostly pediatric, involving multiple DENV serotypes co-circulating and detected through visiting primary health care settings. In addition, our study population was collected from the emergency department of a tertiary hospital during a large-scale outbreak caused by DENV-2 resulting in many severe cases (i.e. a highly selected dengue patients) that might be different from patients infected with other serotypes and from mostly primary health care clinics. Therefore, our results should not be generalized to other settings or dengue-endemic areas. Second, our models aimed to differentiate laboratory-confirmed dengue from non-dengue cases but those cases with missing data or in which laboratory tests were not ordered, or which were regarded as non-dengue cases might be dengue-positive cases due to limitations in the specimens taken or sensitivities of the tests. Third, we had not used confirmed cases of other infectious diseases with similar clinical presentations [44] to verify because of lower case numbers of malaria, chikungunya, other flaviviruses, and rickettsia diseases in Taiwan in recent years. Fourth, our selection of 90% sensitivity and 80% specificity may not be suitable in all epidemiological settings. Besides, in the current model, the machine

was trained to pick up laboratory-confirmed dengue cases from a specific pool of dengue-like illness, which rely on clinicians making a diagnosis of DLI and requesting diagnostic testing of dengue. Therefore, the sensitivity, specificity, and predictive value of our algorithms depend upon the distribution of other clinical diagnoses in the study population. Fifth, patients who came to NCKU around day 1–3 of dengue-like illness might still have high levels of WBCs and platelets before declining, which might lead to misdiagnosis. Since we did not have data on day of illness (fever day) at presentation, it's impossible to know which phases of dengue natural course the patient was at. In fact, many patients came to the emergency department on their first or second day of fever during the 2015 Tainan epidemic. As a result, the cases included in our dataset were likely to be in the early phases, while the cases excluded but with thrombocytopenia were likely to come in later phases of disease through referral or for second opinions. Indeed, we agree fever day is an important feature in dengue diagnosis and management [60]. We are trying to include this information into the entry in our electronic medical record (EMR) system in the near future. Sixth, we investigated only the performance with the DNN, DT, and conventional logistic regression models due to our conjecture that the performance with the prediction models built with other advanced machine learning algorithms such as SVM [29] and random forest [30] will deliver comparable performance. Nevertheless, it would be of interest to investigate whether an aggregated approach may improve prediction performance. Finally, dengue and coronavirus disease 2019 (COVID-19) are difficult to distinguish because they share common clinical and laboratory features [61]. According to the recent data, lymphopenia, fever were common features in COVID-19 patients. Failing to consider COVID-19 because of a positive dengue rapid test result has serious implications not only for the patient but also for public health [62]. Whether the current COVID-19 pandemic caused by a similar pathogen SARS-CoV-2 pose similar difficulties on differential diagnoses is an important concern. In fact, during SARS in Singapore in 2003, overlapping parameters were found for dengue and SARS [63].

Here, we must emphasize that laboratory-confirmation still remains the ultimate method of surveillance and outbreak investigation. Artificial intelligence and other utilities may be helpful when laboratories are overwhelmed. However, it should never replace laboratory-confirmation, even in low and middle income countries. Furthermore, we focused on the prediction power with four key features, without taking into consideration of environmental factors (mosquito indices, female mosquito infection rate, and meteorological factors) that were incorporated in some other studies [64].

Global epidemiology of dengue involves dengue-endemic and non-endemic countries in which the majority of dengue cases are children and adults, respectively. Future efforts require international collaboration, considering levels of endemicity, all four DENV serotypes, areas where vectors of *Aedes aegypti* versus *Aedes albopictus* are the main transmitting DENV vectors, various levels of local resources, types of medical care facilities, population densities, presence of other infectious disease agents with dengue-like clinical presentations, and the scale of epidemic. Most importantly, we sincerely recommend establishing an integrated surveillance and epidemiological informatics, involving clinical, entomological, microbiological/serological, epidemiological, meteorological, and environmental information, as well as measurements of biomarkers important in viral/immuno-pathogenesis of dengue, so that both the magnitude and severity of dengue epidemics can be better predicted. Such integrated surveillance must be community based or even school based [65] for more efficient community mobilization at epidemic sites. In other words, area adjustment using different local data sets to overcome the weaknesses of a certain data set is necessary. This novel approach using machine learning can also extend to other globally important vector-borne infectious diseases [66] to assist in targeting for mosquito control more precisely.

## Supporting information

**S1 Fig. Epidemic curve of the 2015 dengue outbreak in Tainan city and monthly case distribution trend in current study.**
(PDF)

**S2 Fig. A decision tree generated with prior set to 0.388.** This particular tree produced 90.1% sensitivity but only 63.6% specificity. The prediction algorithm traverses the decision tree starting from the root, which is the node at the top of the tree. Each of the branches originating from a node is associated with a criterion of the attribute values. The prediction algorithm moves down along the tree based on the attribute values of the subject for which a prediction is to be made. The "n+" and "n-" symbols in each node respectively denote the number of positive subjects and the number of negative subjects in the training dataset that meet the criteria specified along the path from the root to this particular node. If n+ in a node is larger than n-, then the node is colored by red. Otherwise, the node is colored by blue.
(PDF)

**S3 Fig. Decision tree generated with prior set to 0.636.** This tree produced 66.3% sensitivity and 80.5% specificity. The prediction algorithm traverses the decision tree starting from the root, which is the node at the top of the tree. Each of the branches originating from a node is associated with a criterion of the attribute values. The prediction algorithm moves down along the tree based on the attribute values of the subject for which a prediction is to be made. The "n+" and "n-" symbols in each node respectively denote the number of positive subjects and the number of negative subjects in the training dataset that meet the criteria specified along the path from the root to this particular node. If n+ in a node is larger than n-, then the node is colored by red. Otherwise, the node is colored by blue.
(PDF)

**S1 Table. Comparison of the excluded patients and included cases of ED patients at NCKU Hospital, Jan. 1 to Dec. 31, 2015 in this study.** Pre-peak: Before Epidemic Peak in the Epidemic Curve; SD: Standard Deviation ICU: Intensive Care Units; BP: Blood Pressure; BPM: Heart Rate as Beats per Minute, WBCs: White Blood Cells; CVA: cerebral vascular accident CKD: Chronic Kidney Disease, DM: Diabetes Mellitus.
(PDF)

**S2 Table. Cut-offs employed to stratify numerical variables for building prediction models.**
(PDF)

**S3 Table. Crude Odds Ratios with 95% Confidence Intervals in parentheses.** SBP: Systolic Blood Pressure; DBP: Diastolic Blood Pressure, WBC: White Blood Cells; GCS: Glasgow Coma Scale, CVA: cerebral vascular accident; CKD: Chronic Kidney Disease, DM: Diabetes Mellitus.
(PDF)

**S4 Table. The software packages employed to build the prediction models and the main characteristics of the DNN model.**
(PDF)

**S5 Table. Summary of sensitivities, specificities, Positive Prediction Values (PPVs), and accuracies on subgroup analyses with the three prediction models [Decision Tree (DT), Deep Neural Network (DNN) and Logistic Regression (LR)].** CVA: cerebral vascular accident; CKD: Chronic Kidney Disease, DM: Diabetes Mellitus.
(PDF)

**S6 Table. Subgroup analysis in the Decision Tree (DT) Model.** CVA: cerebral vascular accident; CKD: Chronic Kidney Disease, DM: Diabetes Mellitus.
(PDF)

**S7 Table. Subgroup analysis in the Deep Neural Network (DNN) Model.** CVA: cerebral vascular accident; CKD: Chronic Kidney Disease, DM: Diabetes Mellitus.
(PDF)

**S8 Table. Subgroup analysis in the Logistic Regression (LR) Model.** CVA: cerebral vascular accident; CKD: Chronic Kidney Disease, DM: Diabetes Mellitus.
(PDF)

## Acknowledgments

The authors are grateful for the leadership of Mayor Ching-Te Lai, Dr. Sheng-Zhe Franklin Lin, Dr. Yi Chen, and Dr. Ih-Jen Su in the control of dengue in Tainan since 2015. This study received additional support from the National Mosquito-Borne Diseases Control Research Center of the National Health Research Institutes (NHRI), including Dr. Cheng-Han Lin, Dr. Ya-Fang Wang, Dr. Te-Pin Chang, Ms. Shu-Wen Laura Cheng, Ms. Wen-Ju Lin, and Dr. Ching-Len Liao, whose coordination and support is deeply appreciated. We would also like to sincerely thank all the clinicians, nurses, and laboratory technologists in Tainan hospitals on the front line of medical care for dengue patients and public health professionals in the Tainan City Public Health Bureau for their contributions to the prevention and control of dengue. In addition, we are also greatly appreciative of the feedback provided by several scholars, including Dr. Yee-Shin Lin (Department of Microbiology and Immunology, National Cheng Kung University), Dr. Trai-Ming Yeh (Department of Medical Laboratory Science and Biotechnology, National Cheng Kung University), Dr. K Chang (Department of Internal Medicine, Kaohsiung Medical University Hospital), and Dr. Barry T. Rouse (Genome Science and Technology, University of Tennessee). The English editing efforts by Mrs. Anita Suárez, Mr. Neal Lin, Mr. Nicholas Minahan, and Mr. John Gilbert are also highly appreciated.

## Author Contributions

**Conceptualization:** Tzong-Shiann Ho, Ting-Chia Weng, Jung-Der Wang, Chwan-Chuen King, Yen-Jen Oyang, Ching-Chuan Liu.

**Data curation:** Tzong-Shiann Ho, Ting-Chia Weng, Hsieh-Cheng Han, Hao-Chien Cheng, Chun-Chieh Yang, Chih-Hen Yu, Yen-Jung Liu, Chien Hsiang Hu, Chun-Yu Huang, Ming-Hong Chen.

**Formal analysis:** Hao-Chien Cheng, Chun-Chieh Yang, Chun-Yu Huang.

**Funding acquisition:** Tzong-Shiann Ho.

**Methodology:** Hao-Chien Cheng.

**Project administration:** Hsieh-Cheng Han.

**Resources:** Ching-Chuan Liu.

**Software:** Hao-Chien Cheng, Chien Hsiang Hu, Chun-Yu Huang, Ming-Hong Chen.

**Supervision:** Chwan-Chuen King, Yen-Jen Oyang, Ching-Chuan Liu.

**Validation:** Hao-Chien Cheng, Chun-Chieh Yang, Yen-Jung Liu, Chun-Yu Huang.

**Writing – original draft:** Tzong-Shiann Ho, Ting-Chia Weng, Chih-Hen Yu, Chwan-Chuen King, Yen-Jen Oyang.

**Writing – review & editing:** Tzong-Shiann Ho, Jung-Der Wang, Chwan-Chuen King, Yen-Jen Oyang, Ching-Chuan Liu.

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
