## [Decision Letter · Decision Letter 0]

27 Apr 2020

Dear Dr. Liu,

Thank you very much for submitting your manuscript "Comparing Machine Learning- with Case-Control-Models to Identify Confirmed Dengue Cases" for consideration at PLOS Neglected Tropical Diseases. As with all papers reviewed by the journal, your manuscript was reviewed by members of the editorial board and by several independent reviewers. In light of the reviews (below this email), we would like to invite the resubmission of a significantly-revised version that takes into account the reviewers' comments. 

We cannot make any decision about publication until we have seen the revised manuscript and your response to the reviewers' comments. Your revised manuscript is also likely to be sent to reviewers for further evaluation.

Sincerely,

Alan L Rothman, MD

Associate Editor

Samuel Scarpino

Deputy Editor

Associate Editor comments:

1) The Discussion should be substantially shorter and more focused. As a particular example, the discussion about pathogenesis of leukopenia and thrombocytopenia is unnecessary since these findings are well described and the current study provides no new insights into pathogenesis.

2) The analysis of previous literature is incomplete. The authors need to cite previous papers on prediction models and compare them to the current results. Specific papers not addressed include: Tanner et al. PLoS Negl Trop Dis 2008, Potts et al. PLoS Negl Trop Dis 2010, Park et al. PLoS Negl Trop Dis 2018, and Phakhounthong et al. BMC Pediatr 2018.

3) The authors’ analysis was limited for obvious reasons to DLI, which, as defined, required the clinician’s suspicion of arboviral disease. This represents a potentially significant source of bias, of particular relevance to use of the algorithm for real-time syndromic surveillance as proposed by the authors. The authors should at a minimum acknowledge this limitation and discuss the potential effects of this bias, especially for the comparison of sensitivity and specificity during different time periods. Can the authors address how many cases in the full dataset (>100K visits) during each month would potentially have been identified as suspected dengue by these algorithms?

4) Table S1 shows statistically significant differences between included and excluded groups. The authors should address the potential implications of these differences for the results. Did the authors review whether these subjects could be included in validation of the final models?

5) Line 188- Why were these cutoffs selected? Would other cutoff values have had greater discriminatory value?

6) The WHO definitions (noted in Figure 3) should be described in Methods.

7) Table 3- The clinical relevance of showing results for subgroups based on laboratory data included in the models (e.g., WBC, platelet) is unclear. These subgroup analyses should be deleted.

8) Do the authors have data on day of illness at presentation? If so, these data should be included in Table 1. Did they consider including this variable in the analysis?

9) The timing of the epidemic in Taiwan in 2015 should be noted under Study Population.

10) Line 147- The authors should note the number of visits excluded for incomplete records, cancellation, or re-admission.

11) Lines 298-302- It is unnecessary to repeat data presented in Table 1 in the text.

12) Figure S1 and S2- The labels on each branch should be reformatted to avoid overwriting the graphical representation of the decision tree.

13) References 54 and 57 are identical.

Reviewer's Responses to Questions

**Key Review Criteria Required for Acceptance?**

**Methods**

-Are the objectives of the study clearly articulated with a clear testable hypothesis stated?

-Is the study design appropriate to address the stated objectives?

-Is the population clearly described and appropriate for the hypothesis being tested?

-Is the sample size sufficient to ensure adequate power to address the hypothesis being tested?

-Were correct statistical analysis used to support conclusions?

-Are there concerns about ethical or regulatory requirements being met?

Reviewer #1: Dengue is a major health problem worldwide over the past 50 years. The challenge of identifying dengue cases is increased during large epidemics, especially in non-endemic areas with limited experienced staff. The study group used machine learning (ML)-based prediction models to identify dengue confirmed cases with the rationale of applying these models in health facilities where dengue confirmed tests are not available. 

 It is obvious that the objectives of the study clearly articulated with a clear testable hypothesis stated, and the study design is appropriate to address the stated objectives. 

 Study population consisted of dengue-like illness (DLI) cases (based on ICD9) admitted to the emergency department (ED) from January 1 to December 31, 2015 at National Cheng Kung University Hospital (NCKUH) in Tainan City. The final dataset included 2,942(60.12%) laboratory-confirmed dengue cases and 1,952 non-dengue control cases. The population is clearly described and appropriate for the hypothesis being tested. However, if possible, the authors should present the definition of DLI rather than ICD code. 

 The sample size (totally, 4,894 DLI cases) is sufficient to ensure adequate power to address the hypothesis being tested. However, there is an imbalance of cases between the two groups (confirmed dengue cases and non-dengue control cases) which need to be taken into account when interpreting the results from deep neural network (DNN) models, but it was not mentioned in the manuscript. Moreover, some other questions should be concerned: Is there any repeated variable in data set? And what is the total number of samples (rows) in the data set?

 Correct statistical analysis was used to support conclusion, and there are no concerns about ethical or regulatory requirements being met. About data validation, internal validation with 2 time-repeats was used. As a suggestion, whether it is better if the models are validated with new data set (external validation)?

Reviewer #2: Objectives were clearly formulated, teh study design is appropriate.

**Results**

-Does the analysis presented match the analysis plan?

-Are the results clearly and completely presented?

-Are the figures (Tables, Images) of sufficient quality for clarity?

Reviewer #1: For the results, the analysis presented matches the analysis plan, and the results are clearly and completely presented. The key findings of the study are that using just four input variables [age, body temperature, and counts of white blood cells (WBCs) and platelets], areas under curves (AUCs) of the receiver operating characteristic (ROC) curves were reported at high level (above 80%) for all models, and DNN had higher performance than others.

 Subgroup analyses were informative, all the models were very sensitive particularly in pre-epidemic period. Pre-peak sensitivity (<35 weeks) was 92.6%, 92.9%, and 93.1% in DT, DNN, and LR respectively.

 The figures (tables, images) in the manuscript are of sufficient quality for clarity. It is easier for readers to follow up the paper, S1 Fig and S2 Figure should additionally be explained in text. One point should be considered, the algorithms of the highest performance models were not reported. Could the authors provide them in their manuscript?

Reviewer #2: Results are well presented.

**Conclusions**

-Are the conclusions supported by the data presented?

-Are the limitations of analysis clearly described?

-Do the authors discuss how these data can be helpful to advance our understanding of the topic under study?

-Is public health relevance addressed?

Reviewer #1: The authors’ conclusions are supported by the data presented, and all limitations of the study were clearly reported. Discussion of the manuscript is interesting, and the authors discussed how these data can be helpful to advance our understanding of the topic under study.

 Results of the study reveal that machine-learning based models can be developed to identify dengue cases with four commonly available key features. This implies that the prediction models can be widely deployed to all levels of medical facilities, including hospitals and local clinics. It would be clearer if the authors emphasize more on how these models will be applied in practice.

Reviewer #2: The authors extract a small number of parameters, the combination of which they conclude are predictive of dengue during an outbreak in Taiwan. The identified parameters would also hold true for COVID-19: lymphopenia, fever, etc. 

In fact, during SARS in Singapore in 2003, overlapping parameters were found for dengue and SARS: Clin Infect Dis. 2004 Dec 15;39(12):1818-23. Epub 2004 Nov 19.

Use of simple laboratory features to distinguish the early stage of severe acute respiratory syndrome from dengue fever.

The authors should highlight that laboratory confirmation remains the primary goal of surveillance and outbreak investigation. Artificial intelligence and other parameters may aid when laboratories are overwhelmed, but should never replace laboratory confirmation. in fact, the call is for more enhanced laboratory dengue surveillance in all countries, including low to middle income countries.

**Editorial and Data Presentation Modifications?**

Reviewer #1: The discussion is sometimes rather long for the readers to focus, and should be more concise.

Reviewer #2: none

**Summary and General Comments**

Reviewer #1: The study group used novel machine learning -based prediction models to identify dengue confirmed cases with the rationale of applying these models in health facilities where dengue confirmed tests are not available. As mentioned above, the prediction models to identify dengue cases with four commonly available key features, and can be widely implemented in all levels of medical facilities, and serve as a key component in an integrated dengue surveillance system. Overall, the manuscript is well prepared and organized. I recommend it is considered to be published with minor revision.

Reviewer #2: The authors extract a small number of parameters, the combination of which they conclude are predictive of dengue during an outbreak in Taiwan. The identified parameters would also hold true for COVID-19: lymphopenia, fever, etc. 

In fact, during SARS in Singapore in 2003, overlapping parameters were found for dengue and SARS: Clin Infect Dis. 2004 Dec 15;39(12):1818-23. Epub 2004 Nov 19.

Use of simple laboratory features to distinguish the early stage of severe acute respiratory syndrome from dengue fever.

The authors should highlight that laboratory confirmation remains the primary goal of surveillance and outbreak investigation. Artificial intelligence and other parameters may aid when laboratories are overwhelmed, but should never replace laboratory confirmation. in fact, the call is for more enhanced laboratory dengue surveillance in all countries, including low to middle income countries.

PLOS authors have the option to publish the peer review history of their article (what does this mean?). If published, this will include your full peer review and any attached files.

Reviewer #1: No

Reviewer #2: No
---

## [Editor Report · Decision Letter 1]

18 Aug 2020

Dear Dr. Liu,

Thank you very much for submitting your manuscript "Comparing machine learning- with case-control-models to identify confirmed dengue cases" for consideration at PLOS Neglected Tropical Diseases. As with all papers reviewed by the journal, your manuscript was reviewed by members of the editorial board. The editors appreciated the sincere and significant efforts the authors have made to address the prior review. Based on the response to the prior reviews, we are likely to accept this manuscript for publication, providing that you modify the manuscript according to the additional recommendations below. 

Sincerely,

Alan L Rothman, MD

Associate Editor

Samuel Scarpino

Deputy Editor

We appreciate the significant efforts the authors have made to address the prior review. However, we believe that there are significant issues with this revised manuscript that need to be addressed before it can be considered suitable for publication in the journal:

1) In the previous review, the authors were asked to address potential bias due to the criteria for selection of cases (DLI as defined by ICD-9 codes or request for diagnostic laboratory testing). The authors have provided a detailed tabulation of the number of cases per month, as requested. The authors were also requested to "acknowledge this limitation and discuss the potential effects of this bias"; this point was not addressed. The authors need to modify the Discussion to acknowledge the reliance of their study design on clinicians making a diagnosis of DLI and requesting diagnostic testing. As part of their response, the authors have provided new analyses of excluded cases in the revised manuscript. However, the presentation of these analyses introduces a number of new issues. First, under Results, the section describing their analysis of excluded cases has been placed before the description of the results of the modeling; these sections needs to be re-ordered. Second, the analyses presented in figure S4 and described in the Discussion (lines 592-597) are problematic because they make an assumption that these are true dengue cases and limit consideration to cases with thrombocytopenia who visited the ED after July 2015. It is unsurprising that these steps improve the "accuracy" of the models, but they introduce additional bias into the results; these analyses should be deleted or at least substantially revised. Third, given the distribution of these cases in time relative to the dengue epidemic, it is not surprising that the authors found a high percentage of cases predicted to be dengue. In actuality, it is somewhat surprising that so few non-DLI subjects had complete data for the 4 key parameters. The ready availability of this information in the clinical setting is in fact proposed as a strength of the models. Can the authors comment on what data elements were missing from the other cases?

2) In the previous review, the authors were asked to "highlight that laboratory confirmation remains the primary goal of surveillance and outbreak investigation". The authors have added a very brief acknowledgement of this issue to the Discussion. However, an important limitation of their study is that the sensitivity, specificity, and predictive value of their algorithms depend upon the distribution of other diagnoses in the study population; this should be specifically noted in the Discussion. In addition, a cautionary interpretation note about the need for laboratory confirmation needs to be included in the Abstract and Summary.

3) In the previous review, the authors were asked to cite several additional published papers on prediction models for dengue and "compare them to the current results". The authors have cited these papers in the Introduction but have not addressed the need to compare their results to these other papers in the Discussion.

4) In the previous review, the authors were asked not to repeat data presented in Table 1 in the text (lines 298-302). The authors indicate that this issue was addressed, but their response does not reflect the revised manuscript (lines 334-337).

5) In the previous review, the authors were asked to substantially shorten the Discussion, with specific mention of text discussing mechanisms of leukopenia and thrombocytopenia that are not addressed by the present study. While the Discussion has been reduced in length in this revision, it remains overly long and unfocused; for example, the specific sections noted previously were not deleted.

6) The revised Figures S2 and S3 have misspelled "Negative"; this should be corrected.
---

## [Editor Report · Decision Letter 2]

1 Oct 2020

Dear Dr. Liu,

We are pleased to inform you that your manuscript 'Comparing machine learning- with case-control-models to identify confirmed dengue cases' has been provisionally accepted for publication in PLOS Neglected Tropical Diseases.

Best regards,

Alan L Rothman, MD

Associate Editor

Samuel Scarpino

Deputy Editor

---

## [Editor Report · Acceptance letter]

20 Oct 2020

Dear Dr. Liu,

We are delighted to inform you that your manuscript, "Comparing machine learning with case-control-models to identify confirmed dengue cases," has been formally accepted for publication in PLOS Neglected Tropical Diseases.

Best regards,

Shaden Kamhawi

co-Editor-in-Chief

Paul Brindley

co-Editor-in-Chief
